# Analysis of Metabolites in Gout: A Systematic Review and Meta-Analysis

**DOI:** 10.3390/nu15143143

**Published:** 2023-07-14

**Authors:** Yuanyuan Li, Xu Han, Jinlin Tong, Yuhe Wang, Xin Liu, Zeqi Liao, Miao Jiang, Hongyan Zhao

**Affiliations:** 1Medical Experimental Center, China Academy of Chinese Medical Sciences, Beijing 100700, China; 2Institute of Basic Research in Clinical Medicine, China Academy of Chinese Medical Sciences, Beijing 100700, China

**Keywords:** gout, metabolites, metabolomics, meta-inflammation

## Abstract

(1) Background: Many studies have attempted to explore potential biomarkers for the early detection of gout, but consistent and high levels of evidence are lacking. In this study, metabolomics was used to summarize the changes of metabolites in the literature and explore the potential value of metabolites in predicting the occurrence and development of gout. (2) Methods: We searched the databases including the EMBASE, the Cochrane Library, PubMed, Web of Science, VIP Date, Wanfang Data, and CNKI, and the screening was fulfilled on 30 July 2022. The records were screened according to the inclusion criteria and the risk of bias was assessed. Qualitative analysis was performed for all metabolites, and meta-analysis was performed for metabolite concentrations using random effects to calculate the Std mean difference and 95% confidence interval. (3) Results: A total of 2738 records were identified, 33 studies with 3422 participants were included, and 701 metabolites were identified. The qualitative analysis results showed that compared with the healthy control group, the concentration of 56 metabolites increased, and 22 metabolites decreased. The results of the meta-analysis indicated that 17 metabolites were statistically significant. (4) Conclusions: Metabolites are associated with gout. Some specific metabolites such as uric acid, hypoxanthine, xanthine, KYNA, guanosine, adenosine, creatinine, LB4, and DL-2-Aminoadipic acid have been highlighted in the development of gout.

## 1. Introduction

Gout is a metabolic immune disease characterized by disorders of purine and uric acid (UA) metabolism. It is caused by the super-saturation of UA, which crystallizes in joints, leading to peripheral joint synovitis and severe pain [1,2]. With the increasing number of acute gout attacks, the patient gradually progresses to joint damage, deformity, chronic use-related pain, subcutaneous tophi deposits, and renal damage. Additionally, gout is often associated with metabolic syndrome [3], cerebrovascular diseases [4], cardiovascular diseases [5,6], and renal damage [7]. The prevalence of gout ranges from approximately 1% to 6.8%, with an incidence rate of 0.58 to 2.89 per 1000 persons per year [8]. Moreover, it has doubled in less than 30 years [9]. Studies have shown that gout patients have high healthcare costs due to frequent emergency room visits, hospitalizations, reduced productivity, and disability [10,11,12], which range from USD 172 to USD 6179 [13].

The clinical signs of gout are usually severe pain, edema, redness, and inflammation of the joints, which occur suddenly during the night and then gradually diminish and disappear over a period of days or weeks. Currently, the clinical diagnosis of gout is based on serum uric acid detection, joint synovial fluid examination, and imaging examination [14]. As gout is often acute and easily confused with other forms of arthritis, these diagnostic methods do not seem to be adequately timely and accurate [15], and thus cause patients to suffer from delayed intervention and treatment [16]. UA is a common and important biochemical indicator for the diagnosis of gout, yet not all individuals with hyperuricemia have or will ever develop gout [17,18]. Treatment of gout is divided into acute arthritis treatment and uric acid management. Acute gout is usually treated with high doses of NSAIDs, glucocorticoids, colchicine, and biologic agents over a short period of time. However, NSAIDs may increase the risk of hypertension and acute heart failure [19,20]. Glucocorticoids are associated with potentially serious systemic side effects, including electrolyte disturbances, cardiovascular effects, diabetes, and osteoporosis [21]. Colchicine is widely used in the acute phase of gout [22,23], but the evidence shows that medication adherence to Urate-Lowering Therapy among gout patients was poor worldwide [24,25]. The prevention of acute and chronic gout not only relieves the pain but also reduces associated healthcare costs [26]. Therefore, it is increasingly recognized that early identification and intervention of gout is the most cost-effective path [27].

Gout has a long-term basis of metabolic disorders before the onset. The correlation between metabolism and immunity in the pathogenesis of gout is gradually being noticed [3]. Attention has shifted from simple uric acid metabolites to a variety of other potential metabolites, and exploring changes in metabolites becomes a direction for finding biomarkers [28]. Currently, metabolites have been showing great advantages in disease diagnosis and pathogenesis exploration, providing a new idea for predicting gout [29]. Metabolomics, a platform with powerful spectroscopy and separation techniques, is considered ideal for metabolite determination due to its comprehensiveness and accuracy in metabolite identification [30]. This technique has been used to find biomarkers that predict the development of diseases such as liver cancer, diabetes, osteoarthritis, sepsis, etc. [31,32].

At present, many scholars have attempted to explore the predictive value of metabolites during gout development [33], and some reliable qualitative and quantitative results have been obtained [34]. Some specific metabolites have been confirmed to be related to gout [35,36]. Hypoxanthine and xanthine, metabolic precursors of UA in purine metabolism, have been found to be biomarkers of gout, regardless of whether the patient has hyperuricemia [37,38]. Creatinine clearance, an indicator of kidney function, has been shown to be associated with the early development of subcutaneous urate deposits in gout patients [39]. Methods for the simultaneous measuring of purine, amino acid, creatinine, and other metabolites have been widely developed [40,41]. Thus, metabolites have promising prospects for gout prediction. However, a consistent and comprehensive conclusion is still missing. The aim of this study is to summarize existing studies using metabolomics techniques to detect metabolites in gout patients and to provide comprehensive and reliable conclusions for the prediction of potential biomarkers of gout occurrence.

## 2. Materials and Methods

Systematic reviews and meta-analyses are reported in accordance with the regulations of the Preferred Reporting Program for Systematic Reviews and Meta-analyses (PRISMA-P) [42]. This study is registered in the International Prospective Register of Systematic Reviews (PROPSERO CRD4022366809).

### 2.1. Selection Criteria

The inclusion criteria were: (1) gout patients; (2) the samples (blood, urine, feces, and saliva) from patients and healthy controls were measured and analyzed by metabolomics technique; (3) reported the metabolites profile; (4) cohort study, case–control study, and randomized controlled trial based on human. Exclusion criteria were as follows: (1) obtaining blood samples from patients who are taking gout medication or urate-lowering medication; (2) repeated records; (3) lack of information; (4) and no full text.

### 2.2. Search Strategy

All records were identified by the Cochrane Library, EMBASE, PubMed, Web of Science, VIP Date, Wanfang Data, and CNKI up to 31 July 2022 with the keywords and synonyms combining “gout”, “metabolites”, and “metabolomics” (Appendix A). No restrictions were placed on the language or date of publication.

### 2.3. Study Selection

All identified records were downloaded to Endnote X9, and then the duplicates were removed by computer and manual deletion. Two independent researchers (LYY and TJL) screened the studies by title and abstract, respectively, and then the studies meeting our criteria were found in the full text for further screening. Any disagreements were discussed with the third researcher (TJL) until the team reached a consensus. A senior researcher (ZHY) gave the guidance and supervised.

### 2.4. Risk of Bias Assessment

Two researchers (LYY and HX) assessed the risk of bias in the included records independently by the Newcastle–Ottawa Scale (NOS), a tool for assessing the quality of observational studies [43]. It evaluates research through a “star system” based on selectivity, comparability, and exposure.

### 2.5. Data Collection

The extracted research data included the following items: first author’s name, publication year, country of origin, study design type, language, age, sample size, diagnostic criteria of patients, statistical methods, detection techniques, metabolomics techniques, and trends and concentrations of metabolites in gout and healthy controls. Data were obtained from the article and the Appendix A. When the data is presented graphically only and the original author cannot be contacted for details of the data, we use Web Plot Digitizer (Web Plot Digitizer, V.4.2, San Francisco, California: Ankit Rohatgi, 2019) to extract data from the graph of the article. For those articles with validation cohorts, data from the validation cohorts were also included.

We use PubChem (https://pubchem.ncbi.nlm.nih.gov/ (accessed on 20 January 2023)) to search the metabolites which need to convert their units, and the molecular weight (g/mol) of them are shown in Appendix A. The quartile and median or interquartile range (IQR) were converted into mean and standard deviation (SD). For median and quartile, we tested the skewness first and applied a new piecewise function based on the size of the sample [36,44,45,46,47,48]. For median and IQR, formula 1 was applied. When a study matches two or more experimental groups to one control group, the mean and standard deviation of the experimental group are combined according to formula 2. All formulas we applied were derived from the Cochrane Handbook of Systematic Reviews of Interventions and are shown in Appendix A.

### 2.6. Data Synthesis

A qualitative analysis was performed for changing the direction of metabolites by counting the frequency across the studies. Afterward, the concentrations of metabolites performed the meta-analysis across studies using the standardized mean difference of 46 metabolites (SMD) and confidence intervals of 95% (95% CI). Heterogeneity was deemed substantial if I^2^ was greater than 60%. While in this study, a random effects model was chosen due to the heterogeneity among the records. Funnel plots and Egger’s test were used to assess publication bias when feasible (10 or more studies) [49,50]. Finally, all data synthesis was performed using R software with a meta package (version 3.6.2).

## 3. Results

### 3.1. Literature Search

A total of 2738 records were eventually identified; 902 duplicate records were deleted. Based on the title and abstract screening, 1733 unrelated studies were removed. In the remaining 103 studies, 71 studies were removed for reasons including not providing full text, insufficient data, and failure to meet our criteria. Ultimately, we included 33 studies [28,33,34,35,36,37,44,45,46,47,48,51,52,53,54,55,56,57,58,59,60,61,62,63,64,65,66,67,68,69,70,71,72] and 3422 participants (2149 gout patients and 1273 healthy controls). The flow diagram for the assessment of studies identified in the systematic review is shown in Figure 1.

### 3.2. Characteristics of Included Studies

The included records were published between 1999 and 2021 and all of them were case–control designs, among which 18 studies and 15 studies were reported in English and Chinese, respectively. Participants were from the Slovak Republic, the United States, and China. Age and sex were well matched in all studies, except that 4 studies did not report age, and 7 studies only reported age in the gout group. The studies were classified according to different sample types, including 23 blood samples, 2 urine samples, 2 stool samples, 1 saliva study, and 5 hematuria studies [33,36,47,66,71]. All the included studies measured metabolites mostly using liquid chromatography-mass spectrometry (LC-MS) or gouts chromatography-mass spectrometry (GC-MS) techniques. The specific characteristics of each study are shown in Table 1.

A total of 701 metabolites or their ratios in 33 studies were extracted. Of these, 104 metabolites occur twice or more and can be qualitatively synthesized. The 46 metabolites were distributed across 11 studies with accurate concentration values and units for meta-analysis, among which 9 studies [33,36,37,48,54,67,68,71,72] were directly obtained from the article, and 2 studies [44,46] were extracted by software from the box plots. Median and quartiles were transformed to mean and SD in 3 studies [44,46,69], and median and IQR were transformed in 3 studies [44,46,69]. Six studies [36,44,45,46,47,48] combined the mean and SD. The characteristics of these metabolites are shown in Appendix A.

### 3.3. Risk of Bias of Included Studies

The risk of bias in all included studies was assessed using the NOS scale. The quality of the included studies was generally high with 22 studies (66.7%) receiving over 7 stars. Five studies received 6 stars due to low exposure scores [28,48,51,53,54]. The remaining 4 studies scored less than 6 stars due to inadequate case definition [45,55,56,62]. Results on the risk of bias in the included studies and details of the evaluation of the included studies are presented in Table 1 and Appendix A, respectively.

### 3.4. Qualitative Synthesis

A total of 701 metabolites were identified in 33 studies, of which 567 were in blood samples, 65 were in urine samples, 49 were in fecal samples, and 20 were in saliva samples. Qualitative synthesis was performed on 104 metabolites that were reported more than twice, including 94 blood samples, 5 urine samples, and 1 fecal sample. Based on the sample type classification, 45 metabolites showed increased concentrations, 20 showed decreased concentrations, and 33 showed inconsistent concentrations in blood samples. In urine samples, 2 metabolites showed a decrease in concentration and 3 metabolites showed inconsistent concentrations. In fecal samples, only the concentrations of taurine increased.

In total, 45 metabolites had increased concentrations, 23 had decreased concentrations, and 36 showed inconsistent trends. The details of the qualitative synthesis are given in Table 2. Information about metabolites with inconsistent trends in blood (A) and urine (B) samples is shown in Figure 2.

### 3.5. Meta-Analysis

A total of 46 metabolites were available for meta-analysis, all of them were derived from blood samples. Among these, 17 metabolites were found to be statistically significant by meta-analysis, while 26 were not. The results showed that patients with gout have increased concentrations of UA, hypoxanthine, xanthine, KYNA, LB4, guanosine, 2-Deoxyadenosine, creatinine, 13(S)-HODE, 9(S)-HODE, 5-oxo-ETE, decreased concentration of 12-HETE, 20-carboxy-ARA, 19,20-DHDPA, 11,12-DHET, DL-2-Aminoadipic acid, and adenosine compared to healthy people. However, high-density lipoprotein, low-density lipoprotein, blood urea nitrogen, 11-HETE, 8-HETE, 14(15)EET, 11(12)EET, 8(9)EET, 5(6)EET, 19(20)EDP, 17,18-DHETE, thromboxinB2, inosine, uracil, linoleic acid, 15-HETE, 5-HETE, 13(S)-HOTrE, 9(S)-HOTrE, 13-oxo-ODE, 9-oxo-ODE, 12(13)EpOME, 9(10)EpOME, 12,13-DHOME, 9,10-DHOME, 14,15-DHET, 8,9-DHET, 5,6-DHET, and thromboxin B3 showed no difference between gout and healthy controls. The forest plots for different metabolites between gout and health and the results of the meta-analysis for different metabolites between gout and health are shown in Figure 3 and Table 3.

Effect estimates were relatively consistent across studies, although substantial heterogeneity was found in some metabolite comparisons. We conducted sensitivity analyses for studies with a high risk of bias. However, the results did not change significantly from previous results. At the same time, due to the omissions in the order of individual studies, no study has greater sensitivity.

## 4. Discussion

This paper summarized all studies on metabolite changes in gout patients and identified a characteristic metabolite profile for gout. The meta-analysis found 10 metabolites with significant differences and almost all of them were associated with inflammation.

Since hyperuricemia is the largest risk factor for gout, exploring the production and excretion of UA is fundamental to understanding gout. As the degradation product of purine, the main sources of UA are cell turnover, dietary intake, and de novo synthesis [73], and the primary excretion mode for UA is renal excretion [74,75,76]. Previous studies have shown that decreased renal excretion is the main cause of elevated UA, and high levels of UA are strongly associated with kidney disease [77], yet the mechanism is not clear. We aimed to investigate the role of UA and kidney in gout so that we could find reliable evidence to explain the sequence and pathological mechanism. UA induces vascular lesions characterized by endothelial dysfunction, medial wall thickening, and macrophage infiltration, reducing the blood supply to the renal tubular area and causing ischemic injury in the renal tubular area [7,78]. In addition, studies have shown that the cellular communication between renal tubules and immune cells in the tubulointerstitium plays a key role in the progression of kidney disease [79]. In this process, UA seems to accelerate the interaction between these cells, leading to persistent inflammation and progression of tubulointerstitial fibrosis.

In order to better understand the relationship between UA and the kidney, we need to understand the relationship between the two major systems of metabolism and immunity beforehand. From a survival and evolutionary perspective, the most important ability of a species is to resist starvation and mount an effective immune response to pathogens. To maximize energy efficiency, metabolism, and immunity use the same or overlapping key regulatory molecules and signaling systems [80]. In fact, functional units that control critical metabolic and immune functions in higher organisms evolved from a common ancestral structure, meaning that nutrients can act via the pathogen-sensing system to elicit metabolic or nutrient-induced inflammatory responses. In the long history of human existence, faced with a chronically unstable food supply, the human body has automatically evolved a mechanism to store excess calories. However, with the improvement of human living standards, this once beneficial bond leads to excessive accumulation of nutrients and inflammation in the absence of pathogen infection, this activation is clearly harmful, and the disorder of UA metabolism is one of them. Excessive metabolism produces more UA and activates inflammation, which is the main biological process of kidney disease [81,82,83], and meta-inflammation is the mediator to understanding the close relationship between UA and kidney disease [77,84,85]. In summary, we argue that metabolic excesses of energy and lipids lead to hyperuricemia and activate chronic metabolic inflammation. The excretion of high UA from the kidneys is the result of metabolic inflammation, and UA is both a consequence and a precipitant of renal impairment.

Confirmation of the existence of the same signaling pathway associated with UA and inflammation in gout patients is a prerequisite to proving this conjecture. Our study supports this hypothesis by showing creatinine and DL-2-aminoadipic acid were significantly different between gout and healthy controls. As an intermediate metabolite of lysine metabolism, DL-2-aminoadipic acid plays an important role in the regulation of glucose and lipid metabolism, and studies have shown that DL-2-aminoadipic acid enhances the body’s metabolism and associates with damage to renal function in gout patients [46]. Therefore, the increase in DL-2-aminoadipic acid may act as a compensatory mechanism to promote the body to better carry out excess energy and fat consumption. Hypoxanthine has been shown to have the potential to cause renal inflammation in addition to its various pathophysiological consequences in humans [86,87]. As a purine metabolite, adenosine acts primarily on G protein-coupled receptors and plays an anti-inflammatory role in regulating the onset and remission of gout [88,89]. Specifically, it binds to different purinergic receptors to regulate IL-1β secretion, which is involved in the pathogenesis of gout attack [88]. Kynurenic acid (KYNA), as a branch product of the tryptophan metabolism pathway, is involved in the regulation of the cells of the immune system and a variety of immune-mediated diseases [90]. According to the newest research, chronic stress or mild inflammation can promote the production of KYNA and promote various immunomodulatory actions due to KYNA-mediated signaling pathways [91].

Having explored the meta-inflammatory basis of gout flares, we then turn to inflammation in the conventional sense caused by sodium urate crystals. In a vicious cycle of metabolic surplus and inflammation, eventually crystals precipitate and gout arthritis occurs. Sodium urate can promote the secretion of pro-inflammatory cytokines, which act on the kidneys, leading to a further decline in kidney function [66,83]. At the same time, monosodium urate crystals precipitate in the less-perfused joints and then induce an inflammatory response in the immune system [92,93]. Studies by RAE SA et al. have shown that LB4 is an important chemical mediator in acute gout attacks [94]. LB4 stimulates the production of multiple pro-inflammatory cytokines and mediators, and pharmacological data suggest that it could enhance and prolong tissue inflammation [95].

Metabolic inflammation and inflammation induced by sodium urate crystals are the basis for understanding the pathology of gout. Our study also confirmed the importance of inflammation in the pathogenesis of gout. The present study suggests that the signaling pathways used by metabolic inflammation and traditional inflammation are relatively consistent [96], starting from the root of energy metabolism and finding related biomarkers. It may be used as a new idea for early detection and prevention of gout attacks.

## 5. Limitations of the Study

Our study does have some limitations. (1) The main limitation of the study is that, while the direction of effect estimates is relatively consistent, the results of the meta-analysis do have a high degree of heterogeneity. Heterogeneity is inevitable due to different study populations, different body conditions, and different methods of metabolite detection. (2) Although we use a new method of partition functions in converting sample sizes to means and standard deviations, the distribution of the data is still problematic. In further studies, we hope to better address this issue. (3) There was no search of the clinical trial registry database, which may have resulted in an inadequate comprehensive search.

## 6. Conclusions

Changes in metabolites in humans do shed light on the underlying mechanisms of gout development. Some specific metabolites such as UA, hypoxanthine, xanthine, KYNA, guanosine, adenosine, creatinine, LB4, DL-2-Aminoadipic acid have been highlighted in the development of gout, establishing the potential of metabolites as predictive biomarkers of gout. However, these findings require further investigation and validation in larger prospective cohort studies.

## Figures and Tables

**Figure 1 nutrients-15-03143-f001:**
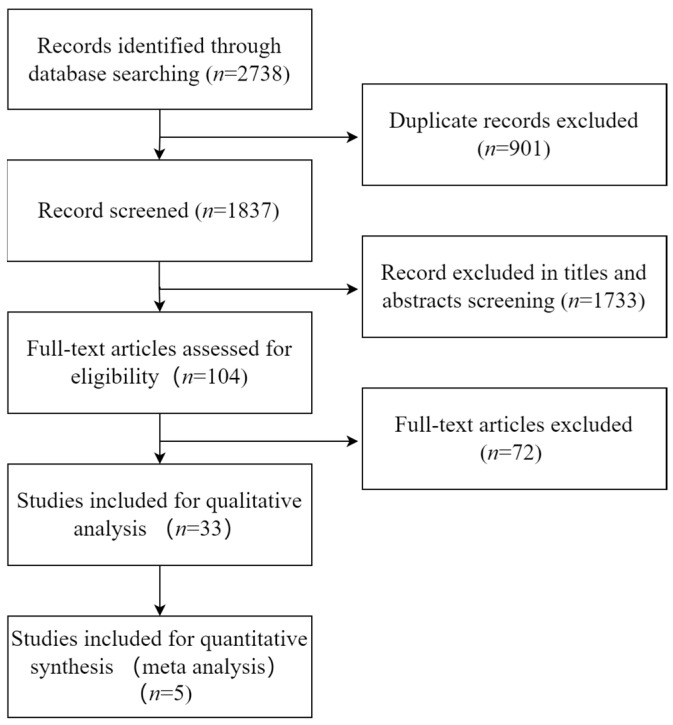
Flow diagram for the assessment of studies identified in the systematic review.

**Figure 2 nutrients-15-03143-f002:**
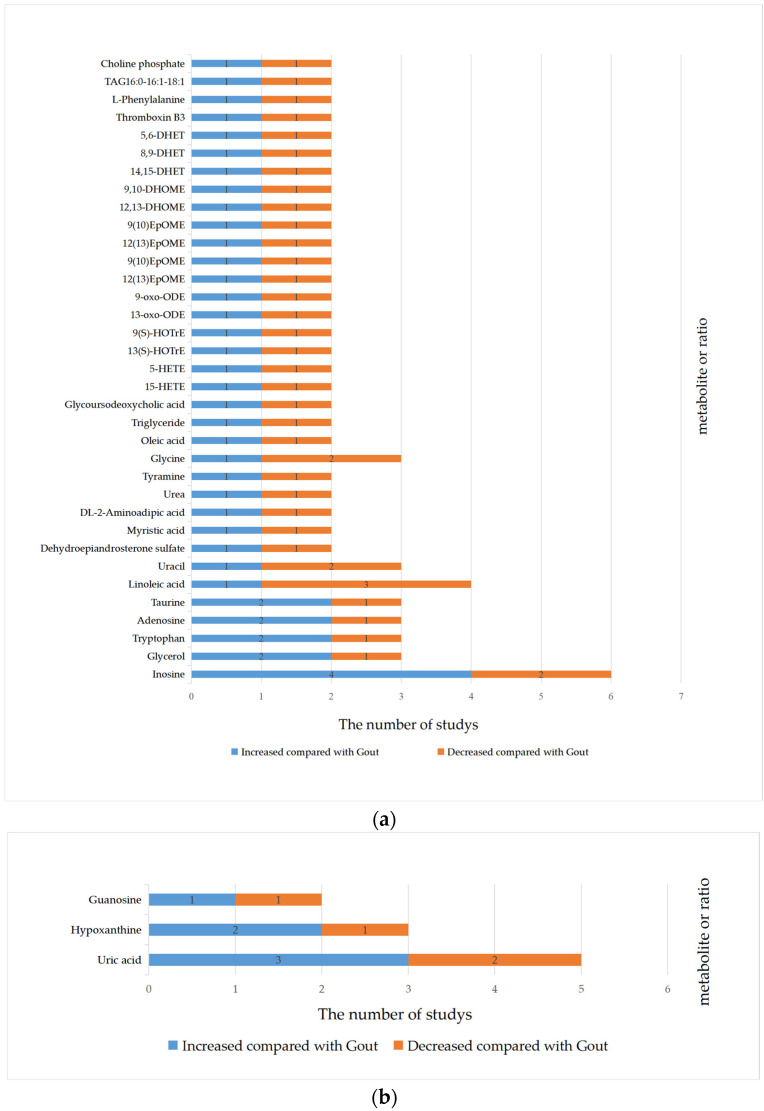
The occurrence frequency of metabolites with different concentration trends between gout and health: (**a**) blood sample; (**b**) urine sample.

**Figure 3 nutrients-15-03143-f003:**
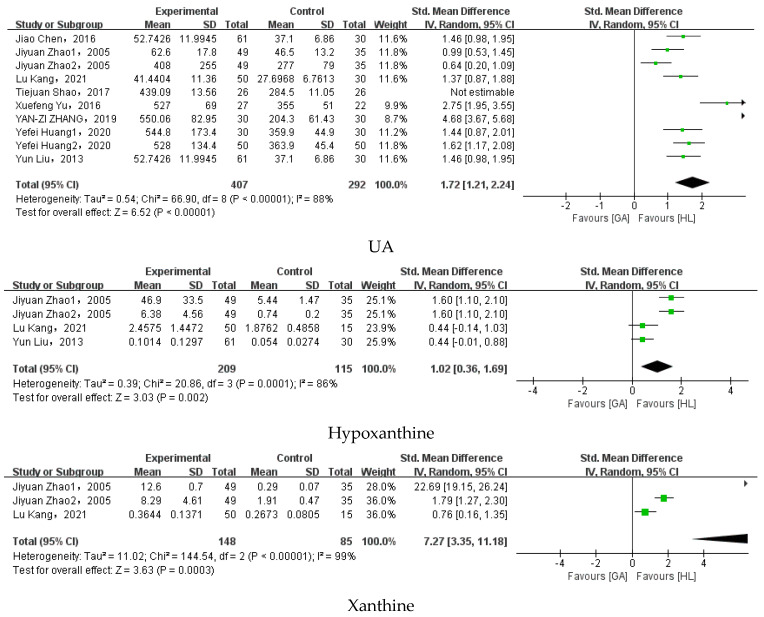
Forest plot for different metabolites between gout and health. Data from references [7,28,33,34,35,36,37,44,45,46,47,48,51,52,53,54,55,56,58,59,60,61,62,63,64,65,66,67,68,69,70,71,72].

**Table 1 nutrients-15-03143-t001:** The characteristics and NOS rate of the included studies.

Author (Year)	Country	Sample (Gout/Health)	Age (Gout/Health)	Metabolomics Technique	NOS
Fanghui Qiu et al. (2018) [51]	China	10/10	46.70 ± 8.69/NG	LC-MS	6
Tingting Yin et al. (2013) [52]	China	29/22	45.6 ± 13.0/45.7 ± 12.2	LC-GC	7
Zheng Zhong et al. (2020) [35]	China	31/31	34.72 ± 10.62/34.83 ± 8.16	UPLC-Q-TOF/MS	7
Yitao Li et al. (2019) [55]	China	34/60	51 ± 12/26 ± 12	LC-GC	5
Shang Lv et al. (2020) [45]	China	69/80	49.23 ± 17.90/NG	UPLC-Hqtof-MS	8
Tie Zhao et al. (2013) [56]	China	29/22	45.6 ± 13.0/45.7 ± 12.2	UPLC-Q-TOF/MS	5
Meijiao Wang et al. (2013) [58]	China	53/40	33.1 ± 8.6/30.6 ± 6.6	NMR	7
Yun Liu et al. (2013) [36]	China	21/20	48.3 ± 16.7/44.8 ± 13.12	HPLC	7
Jiao Chen et al. (2016) [48]	China	29/26	50.3 ± 11.4/NG	GC-MS	6
Mingmei Zhang et al. (2021) [59]	China	50/50	38.61 ± 12.6/NG	UPLC-MS	7
Xuefeng Yu et al. (2016) [54]	China	20/15	NG	UPLC	6
Jiyuan Zhao et al. (2005) [37]	China	49/35	NG	HPLC-MS-MS	4
Tingting Yin et al. (2021) [53]	China	29/22	45.6/18–70	UPLC/MS/MS	6
Fanshu Sun et al. (2019) [7]	China	57/92	48.26 ± 14.21/46.85 ± 12.50	UPLC-MS	7
Qilin Huang et al. (2014) [47]	China	60/30	44. 8 ± 8.2/NG	GC-MS	7
Tiejuan Shao et al. (2017) [68]	China	26/26	43.60 ± 1.98/39.42 ± 2.33	1H NMR	7
Yuqi Chen et al. (2021) [60]	China	58/20	43 ± 15.68/43.0 ± 8.6	GC-TOF-MS	7
Li Cui et al. (2017) [61]	China	8/15	NG	MS/MS/MS	7
Yefei Huang et al. (2020) [33]	China	30/30	44.27 ± 12.69/39.93 ± 9.57	UPLC-MS	7
Kang Lu et al. (2021) [44]	China	50/15	NG	UPLC	3
Yun Liu et al. (2011) [66]	China	21/20	48.3 ± 16.70/47.4 ± 14.24	HPLC-DAD	6
Ying Luo et al. (2019) [67]	China	26/26	48.9 (12.8)/51.3 (9.3)	LC-MS/MS	7
Yun Liu et al. (2012) [28]	China	45/41	43 (20–74)	HPLC-DAD	7
Shang Lyu et al. (2022) [46]	China	295/80	46.5 ± 15.8/52.1 ± 9.3	UHPLC-QTOF-MS-MS	8
Yannan Zhang et al. (2018) [70]	China	49/50	45.6 ± 7.3/43.8 ± 11.5	H NMR	7
Lisa K. Stamp et al. (2014) [69]	New Zealand	31/27	60.6 (40~91)/58.1 (39~79)	HPLC	7
Zizhang Yan et al. (2019) [71]	China	30/30	49.56 ± 11.78/44.32 ± 11.51	UPLC-Q-TOF-MS	4
Jiyuan Zhao et al. (2005) [72]	China	12/35	NG	HPLC-UV-MS/MS	8
Shijia Liu et al. (2022) [65]	China	183/88	51.3 ± 13.8/46.3 ± 15.8	UHPLC-Q	8
Jiang Miao et al. (2013) [63]	China	33/60	51 (30–69)/34 (25–74)	(GC−TOF MS) and (UPLC−QTOF MS)	8
Shen Xia et al. (2021) [34]	China	109/119	43.94 ± 11.88/46.77 ± 10.14	LC-MS	8
Richard et al. (1999) [62]	Slovak	28/18	50.2 ± 10.3	HPLC	4
Qianqian Li et al. (2018) [64]	China	35/29	45.3 ± 1.8/43.1 ± 1.6	GC-MS	8

Notes: LC-MS: liquid chromatography-mass spectrometry; LC-GC: liquid chromatography-gas chromatography; UPLC-Q-TOF/MS: ultra performance liquid chromatography quadrupole time-of-flight mass spectrometry; UPLC-Hqtof-MS: ultra performance liquid chromatography quadrupole time-of-flight mass spectrometry; NMR: nuclear magnetic resonance imaging; HPLC: high performance liquid chromatography; GC-MS: gas chromatograohy mass spectrometry; UPLC-MS: ultra performance liquid chromatography/tandem mass spectrometry; UPLC: ultra performance liquid chromatography; HPLC-MS-MS: high performance liquid chromatography–tandem mass spectrometry; UPLC/MS/MS: ultra performance liquid chromatography/tandem mass spectrometry; 1H NMR: nuclear magnetic resonance; GC-TOF-MS: gas chromatography time-of-flight mass spectrometry; HPLC-DAD: high performance liquid chromatography with diode array detector; UHPLC-QTOF-MS-MS: ultra performance liquid chromatography quadrupole time-of-flight mass spectrometry; GC−TOF MS: gas chromatography–time-of-flight mass spectrometry.

**Table 2 nutrients-15-03143-t002:** The details of the qualitative synthesis.

Concentration Trend	Small Molecule Metabolite Name		
	Blood Samples	Urine Samples	Fecal Samples
Upward	Uric acid, Phenylalanine, Hypoxanthine, Xanthine, Creatinine, Kynurenic acid, Mannose, Mannitol, Leukotriene B4, Leucine, Guanosine, Gluconic acid, Creatine, 13(S)-HODE, 2-deoxyadenosine, 2PY, 5-oxo-ETE, 9(S)-HODE, Acetylornithine, Alanine, Arabitol, Aspartate, Aspartic acid, Blood urea nitrogen, Cis-5,8,11,14,17-eicosapentaenoic acid, Cysteine, D-Gluconic acid, Dihydroxyfumaric acid, Glyceraldehyde, Homoserine, Indoleacetic acid, Isoleucine, Lactic acid, L-Ornithine, Low density lipoprotein, LPC14:0, LPC20:3, LPE18:0, LysoPC(16:0), Malic acid, PE16:0-18:2, PE18:0-18:1, Succinic acid, Thromboxin B2, Valine		Taurine
Downward	Arachidonic acid, LysoPC(18:2(9Z,12Z)), Lauric acid, Threonate, Stearic acid, High-density lipoprotein, 11-HETE, 8-HETE, 20-carboxy-ARA, 14(15)EET, 11(12)EET, 8(9)EET, 5(6)EET, 19(20)EDP, 19,20-DHDPA, 17,18-DHETE, 11,12-DHET, TAG18:0-18:1-22:1, TAG18:1-20:0-22:1, 12-HETE	Tryptophan, Creatinine	
Inconsistent	Inosine, Linoleic acid, Glycerol, Uracil, Tryptophan, Adenosine, Taurine, Dehydroepiandrosterone sulfate, Myristic acid, DL-2-Aminoadipic acid, Urea, Tyramine, Glycine, Oleic acid, Triglyceride, Glycoursodeoxycholic acid, 15-HETE, 5-HETE, 13(S)-HOTrE, 9(S)-HOTrE, 13-oxo-ODE, 9-oxo-ODE, 12(13)EpOME, 9(10)EpOME, 12(13)EpOME, 9(10)EpOME, 12,13-DHOME, 9,10-DHOME, 14,15-DHET, 8,9-DHET, 5,6-DHET, Thromboxin B3, L-Phenylalanine, TAG16:0-16:1-18:1, Choline phosphate	Uric acid, Hypoxanthine, Guanosine	

Notes: HODE: hydroxyoctadecadienoic acid; 2PY: N1-Methyl-2-pyridone-5-carboxamide; oxo-ETE: oxo-eicosatetraenoic acid; LPC: lysophosphatidylcholine; LPE: lysophosphatidylethanolamine; LysoPC: lysophosphatidylcholine; PE: phosphatidylethanolamine; HETE: hydroxyeicosatetrasanoic acid; EET: epoxyeicosatrienoic acid; EDP: epoxydocosapentaenoic acid; DHDPA: dihydroxydocosapentaenoic acid; DHETE: dihydroxyeicosateteaenoic acid; TAG: triacylglycerol; HOTrE: hydroxyoctadecadienoic acid; oxo-ODE: oxo-octadecadienoic acid; EpOME: epoxyoctadecamonoeneoic acid; DHOME: dihydroxyoctadecamonoeneoic acid; DHET: dihydroxyeicosatrienoic acid. The following articles and tables are the same.

**Table 3 nutrients-15-03143-t003:** The results of meta-analysis for different metabolites between gout and health.

Small Molecule Metabolites	Studies for Synthesis	SMD	95% CI	I^2^	*p*-Value
Uric acid	Jiyuan Zhao et al. (2005) [37]; Jiyuan Zhao et al. (2005) [72]; Yun Liu et al. (2013) [36]; Jiao Chen et al. (2016) [48]; Xuefeng Yu et al. (2016) [54]; Tiejuan Shao et al. (2017) [68]; Zizhang Yan et al. (2019) [71]; Yefei Huang et al. (2020) [33]; Kang Lu et al. (2021) [44]	2.27	[1.55, 2.99]	93%	*p* < 0.00001
Kynurenic acid	Shang Lv et al. (2020) [45]; Shang Lyu et al. (2022) [46]	0.58	0.36–0.79	0%	*p* < 0.00001
Guanosine	Jiyuan Zhao et al. (2005) [37]; Jiyuan Zhao et al. (2005) [72]	0.9	[0.58, 1.23]	0%	*p* < 0.00001
Creatinine	Tiejuan Shao et al. (2017) [68]; Zizhang Yan et al. (2019) [71]; Yefei Huang et al. (2020) [33]; Kang Lu et al. (2021) [44]	1.4	[0.96, 1.84]	67%	*p* < 0.00001
DL-2-Aminoadipic acid	Shang Lv et al. (2020) [45]; Shang Lyu et al. (2022) [46]	1.45	[1.21, 1.69]	0%	*p* < 0.00001
Adenosine	Jiyuan Zhao et al. (2005) [37]; Jiyuan Zhao et al. (2005) [72]; Yun Liu et al. (2013) [36]	1.17	[0.89, 1.44]	0%	*p* < 0.00001
19,20-DHDPA	Ying Luo et al. (2019) [67]	−0.92	[−1.35, −0.49]	0%	*p* < 0.0001
Xanthine	Jiyuan Zhao et al. (2005) [37]; Jiyuan Zhao et al. (2005) [72]	7.27	[3.35–11.8]	99%	*p* = 0.0003
5-oxo-ETE	Ying Luo et al. (2019) [67]	0.57	[0.15, 0.99]	0%	*p* = 0.008
Leukotriene B4	Ying Luo et al. (2019) [67]	0.59	[0.17, 1.01]	0%	*p* = 0.005
Hypoxanthine	Jiyuan Zhao et al. (2005) [37]; Jiyuan Zhao et al. (2005) [72]; Yun Liu et al. (2013) [36]; Kang Lu et al. (2021) [44]	1.02	[0.36, 1.69]	86%	*p* = 0.002
2-Deoxyadenosine	Jiyuan Zhao et al. (2005) [37]; Jiyuan Zhao et al. (2005) [72]	0.38	[0.07, 0.69]	0%	*p* = 0.02
13(S)-HODE	Ying Luo et al. (2019) [67]	0.51	[0.09, 0.92]	0%	*p* = 0.02
9(S)-HODE	Ying Luo et al. (2019) [67]	0.52	[0.10, 0.93]	0%	*p* = 0.02
11,12-DHET	Ying Luo et al. (2019) [67]	−0.5	[−0.91, −0.08]	0%	*p* = 0.02
12-HETE	Ying Luo et al. (2019) [67]	−0.73	[−1.38, −0.08]	56%	*p* = 0.03
20-carboxy-ARA	Ying Luo et al. (2019) [67]	−0.46	[−0.87, −0.04]	0%	*p* = 0.03
High-density lipoprotein	Xuefeng Yu et al. (2016) [54]; Zizhang Yan et al. (2019) [71]	−1.28	[−3.05, 0.48]	93%	*p* = 0.15
Low-density lipoprotein	Xuefeng Yu et al. (2016) [54]; Zizhang Yan et al. (2019) [71]	2.37	[−0.02, 4.75]	95%	*p* = 0.05
Blood urea nitrogen	Tiejuan Shao et al. (2017) [68]; Zizhang Yan et al. (2019) [71]	2.47	[−0.63, 5.57]	97%	*p* = 0.12
11-HETE	Ying Luo et al. (2019) [67]	−0.49	[−1.18, 0.20]	62%	*p* = 0.16
8-HETE	Ying Luo et al. (2019) [67]	−0.2	[−0.61, 0.21]	0%	*p* = 0.34
14(15)EET	Ying Luo et al. (2019) [67]	−0.93	[−2.07, 0.22]	85%	*p* = 0.11
11(12)EET	Ying Luo et al. (2019) [67]	−0.84	[−2.07, 0.39]	87%	*p* = 0.18
8(9)EET	Ying Luo et al. (2019) [67]	0.91	[−1.95, 0.13]	82%	*p* = 0.09
5(6)EET	Ying Luo et al. (2019) [67]	−1.04	[−2.53, 0.45]	9%	*p* = 0.17
19(20)EDP	Ying Luo et al. (2019) [67]	−0.59	[−1.47, 0.28]	76%	*p* = 0.18
17,18- DHETE	Ying Luo et al. (2019) [67]	−0.42	[−0.83, −0.00]	0%	*p* = 0.05
Thromboxin B2	Ying Luo et al. (2019) [67]	0.25	[−0.16, 0.66]	0%	*p* = 0.23
Inosine	Jiyuan Zhao et al. (2005) [37]; Jiyuan Zhao et al. (2005) [72]; Yun Liu et al. (2013) [36]; Kang Lu et al. (2021) [44]	0.09	[−1.07, 1.26]	95%	*p* = 0.87
Uracil	Yun Liu et al. (2013) [36]; Shang Lyu et al. (2022) [46]	−5.14	[−15.12, 4.84]	99%	*p* = 0.31
Linoleic acid	Qilin Huang et al. (2014) [47]; Xuefeng Yu et al. (2016) [54]	−0.36	[−3.69, 2.97]	98%	*p* = 0.83
15-HETE	Ying Luo et al. (2019) [67]	−0.27	[−0.91, 0.37]	57%	*p* = 0.40
5-HETE	Ying Luo et al. (2019) [67]	−0.28	[−2.37, 1.80]	85%	*p* = 0.79
13(S)-HOTrE	Ying Luo et al. (2019) [67]	0.95	[−0.15, 2.05]	66%	*p* = 0.09
9(S)-HOTrE	Ying Luo et al. (2019) [67]	0.42	[−0.65, 1.48]	84%	*p* = 0.44
13-oxo-ODE	Ying Luo et al. (2019) [67]	−0.31	[−2.55, 1.94]	96%	*p* = 0.79
9-oxo-ODE	Ying Luo et al. (2019) [67]	−0.15	[−2.28, 1.97]	96%	*p* = 0.89
12(13)EpOME	Ying Luo et al. (2019) [67]	0.47	[−0.75, 1.70]	88%	*p* = 0.45
9(10)EpOME	Ying Luo et al. (2019) [67]	0.06	[−0.92, 1.05]	82%	*p* = 0.90
12,13-DHOME	Ying Luo et al. (2019) [67]	0.26	[−0.63, 1.15]	78%	*p* = 0.57
9,10-DHOME	Ying Luo et al. (2019) [67]	0.00	[−1.09, 1.09]	85%	*p* = 1.00
14,15-DHET	Ying Luo et al. (2019) [67]	−0.04	[−1.07, 0.98]	83%	*p* = 0.93
8,9-DHET	Ying Luo et al. (2019) [67]	0.42	[−0.63, 1.47]	83%	*p* = 0.44
5,6-DHET	Ying Luo et al. (2019) [67]	−0.47	[−2.42, 1.49]	95%	*p* = 0.64
Thromboxin B3	Ying Luo et al. (2019) [67]	0.25	[−0.41, 0.41]	0%	*p* = 0.98

## Data Availability

The data reported in this review are all freely available through a database search using the included search terms.

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
