# Peer review of "Analysis of Metabolites in Gout: A Systematic Review and Meta-Analysis"

_nutrients, 2023, doi:10.3390/nu15143143_

Round 1

Reviewer 1 Report

An observational study of this nature is difficult at best and the authors state this in the limitations. However, in trying to define a metabolic profile of a disease like gout, consideration of factors such as sex, metabolic issues such as menopause status, relation of gouty attack and hyperuricemia,,as well as comorbidities  specifically T2DM and lipid levels, presence or absence of kidney disease, hypertension and obesity, and CAD, need to be considered.  . Also medications need to be addressed. Statistical analysis should be applied that consider these variables.  The metabolic findings would be easier to understand if grouped by metabolic pathways and/or functional pathways such as inflammation/inflammasome, hyperuicemia and acute gout. Other metabolic pathways including glucose metabolism and T2DM etc are probably related to increased uric acid and gout, but how? Why are the forrest plots capturing congruency of authors; they could be better used to demonstrate significance of clinical variables. How do the predictive markers mentioned in the conclusion result in gout and or hyperuricemia?

The tables 1 and 2 should be in the suppl.

Number of syntax errors in methods section 

Author Response

请参阅附件。

Reviewer 2 Report

article section

error

description

1

all

English quality

Minor revision by an expert is needed.

2

Introduction

abbreviation management

“Gout” is only 1 word, 4 letters, you do not need to abbreviate it to “GA”. It is contra-intuitive and it makes the text harder to read.

3

Introduction

academic style

“Rock deposition” - “rock” is not an appropriate term, it is “tophi”.

4

Introduction

scientific fact

“The clinical diagnosis of GA is based on serum uric acid detection, joint synovial fluid examination and imaging examination”. The authors forgot to mention clinical manifestations (e.g., anamnesis, arthritis). Please add them.

5

Introduction

fact omission

“It has been found that not all GA patients have elevated UA”. In fact, not all persons with hyperuricemia have or will ever develop gout. This is worth mentioning.

6

Introduction

information redundancy

“bone mineral density, and osteoporosis”. Osteoporosis can be diagnosed by revealing low bone mineral density, therefore it suffices to write “osteoporosis”.

7

Introduction

academic style

“is not friendly to patients”. Please use a more impersonal rigorous style. For example: “colchicine is associated with higher risk of kidney damage”.

8

Introduction

incomprehensibility

“early development of subcutaneous routes in GA patients”. What are “subcutaneous routes”? Do you mean tophi? Please clarify and update the text accordingly.

9

Data synthesis

abbreviation management

“Std mean difference” – please write the full name: “standardized mean difference”

10

Characteristics of included studies

academic style

“46 metabolites with accurate concentration”. Please do not start a phrase with a numeral.

11

Qualitative synthesis

academic style

“2 metabolites showed a decrease in concentration”. Please do not start a phrase with a numeral.

12

Characteristics of included studies

reference management

“All GA patients were diagnosed according to the clinical diagnostic guidelines.” Please indicate what guidelines and include their reference.

13

Table 1

incomprehensibility

The footnotes contain the phrase “The following articles and tables are the same”. What does that mean? If it is not important, please remove it.

14

Table 1

abbreviation management

The third column name is “Sample (GA/HL)”. “GA” is “gout”, but “HL” is explained in the footnote as “health”. Do you mean “healthy controls”? Please clarify and update the text accordingly.

15

Table 1

abbreviation management

Please explain in the footnote all the other abbreviations used in the table, for example NG, LC-MS, LG-GC etc. A table should be understandable alone from the text.

16

Table 2

incomprehensibility

Please do not use “/” to indicate the absence of metabolites, since this is misleading the reader. Just leave the cell blank.

17

Figure 3

typing error

Figure 3 (Forest plots) is wrongly numbered as Figure 2.

18

Meta-analysis

information redundancy

The text of subchapter “3.5. Meta-analysis” is unreadable because of all the numbers it reports. These numbers are also found in figure 3 and table 3, so please clean the text.

19

Discussion

incomprehensibility

“meta-inflammation” – what does this term mean? Please clarify in the text.

20

Discussion

incomprehensibility

“hyper UA”. This is not an abbreviation and not an actual term. Please use “hyperuricemia”.

21

Discussion

scientific fact

“pro-inflammatory cytokines, which are deposited in the kidney”. In fact, cytokines are not deposited, but they act on the kidney.

22

Discussion

abbreviation management

“MSU crystals”. This abbreviation is not explained anywhere in the text.

23

Discussion

scientific fact

“MSU crystals precipitate in the less perfused joints and then trigger the anti-inflammatory response of the immune system”. MSU crystals induce an inflammatory response, not an anti-inflammatory response.

24

Discussion

causality

“The journal 'The Lancet' revealed that LB4 is a significant chemical mediator during acute GA attacks”. It is not Lancet who revealed this, but the authors of study reference 94 who did, so there is no point in naming the journal.

25

Limitations of the study

incomprehensibility

“gray literature”. What is this? Please clarify or rephrase.

Minor English revision by an expert is needed.

Author Response

请参阅附件。
